# Sugar-Sweetened Beverage Consumption and Associated Health Risks Awareness Among University Students in Kuwait: A Cross-Sectional Study

**DOI:** 10.3390/nu17101646

**Published:** 2025-05-12

**Authors:** Tasleem A. Zafar, Dalal U. Z. Alkazemi, Hasan Muthafar, Hommam Alanzi, Jiwan S. Sidhu

**Affiliations:** 1Department of Food Science and Nutrition, College of Life Sciences, Kuwait University, P.O. Box 5969, Safat 13060, Shadadiya, Kuwait; dalal.alkazemi@ku.edu.kw (D.U.Z.A.); hasan.muthafar@grad.ku.edu.kw (H.M.); 2Faculty of Science, University of Ottawa, Ottawa, ON K1N 6N5, Canada; halan089@uottawa.ca; 3Department of Livestock Products Technology, Guru Angad Dev Veterinary and Animal Sciences University, Ludhiana 141012, India; dr.jiwan.sidhu@gmail.com

**Keywords:** sugar-sweetened beverages, sugar content, university students, awareness, disease risks, Kuwait

## Abstract

**Background:** Excessive consumption of sugar-sweetened beverages (SSBs) is linked to various health risks, including obesity, type 2 diabetes, and cardiovascular diseases. University students are particularly vulnerable due to lifestyle factors and high consumption patterns. **Objective:** This cross-sectional survey examined SSB consumption patterns, sugar intake, and awareness of health risks among Kuwait University students. **Methods:** Statistical analyses, including chi-square tests and logistic regression models, were conducted using SPSS. **Results:** Findings revealed a high prevalence of SSB consumption, with significant associations between intake levels and demographic characteristics. Regular soda was the most consumed SSB, with 42% of students drinking it 5–6 times per week and 32% consuming it daily. The median daily sugar intake from soda alone was 38 g, approaching or exceeding recommended limits. Overall, 34% of students were classified as high-sugar consumers. Males had a higher total sugar intake, while females consumed SSBs more frequently. Greater health awareness was associated with lower sugar consumption, such as obesity (OR = 0.142, 95% CI = 0.046–0.435, *p* < 0.001), whereas students who were aware of the sugar content in SSBs and who preferred unsweetened fruit juices had significantly lower sugar intake from SSBs (OR = 0.653, 95% CI = 0.435–0.980, *p* = 0.040; OR = 0.447, 95% CI = 0.295, 0.675; *p* < 0.001, respectively). **Conclusions**: The findings underscore the urgent need for targeted interventions—such as educational campaigns and policy measures—to reduce SSB consumption and promote healthier dietary habits among young adults in Kuwait.

## 1. Introduction

The increasing prevalence of obesity and non-communicable diseases (NCDs) poses a significant public health challenge worldwide, particularly in Kuwait. A major contributor to these conditions is the excessive consumption of sugar-sweetened beverages (SSBs), which include drinks sweetened with added sugars such as white or brown sugar, high-fructose corn syrup, dextrose, and others. These beverages include sodas, sports drinks, energy drinks, and sweetened iced teas. According to the Centers for Disease Control and Prevention, SSBs are a primary source of added sugars in the diet, providing empty calories that contribute to poor dietary quality and increased risk of health issues [1]. Research highlights a strong association between high SSB consumption and adverse health outcomes, such as obesity, type 2 diabetes, cardiovascular diseases, and tooth decay [2,3,4]. Unlike solid foods, liquid calories from SSBs are less satiating and are often not compensated for by reduced intake at subsequent meals, leading to an overall increase in daily caloric intake [5,6].

Kuwait has one of the highest obesity rates globally, with approximately 80% of adults classified as overweight or obese and 23% diagnosed with type 2 diabetes [7,8]. Among adolescents, nearly 50% are overweight or obese [9], underscoring the urgency of addressing dietary habits, including SSB consumption. Reports indicate that 99% of the Kuwaiti population consumes at least one cup of carbonated beverages daily, often exceeding recommended sugar intake limits [10,11]. Despite extensive global research on SSB consumption, limited studies have quantified sugar intake from these beverages in Kuwait, particularly among university students.

University students are a key demographic for dietary interventions, as they establish lifelong eating habits during this growing stage. Understanding their SSB consumption patterns and awareness levels is crucial for designing targeted strategies to reduce intake and encourage healthier choices. This study aims to assess the prevalence and patterns of SSB consumption among Kuwait University students, quantify their daily sugar intake from these beverages, and evaluate their awareness of the associated health risks. By identifying demographic and behavioral factors influencing SSB consumption, the findings of the study can inform public health strategies to mitigate the rising burden of diet-related diseases in Kuwait. This study is among the first to quantify daily sugar intake from SSBs and assess health risk awareness specifically among university students in Kuwait, addressing a critical gap in regional research and offering targeted insights for public health interventions.

## 2. Materials and Methods

### 2.1. Study Design and Population

This cross-sectional study was conducted at Kuwait University, targeting students aged 18 years and above. Data collection occurred between May and July 2022. Ethical approval for the study was obtained from the Ethical Review Committee at the College of Life Sciences, Kuwait University, under the code KU-CLS-22-04-17. Informed consent was obtained from all participants before survey administration. Participants were not offered any financial or material incentives for completing the survey; participation was entirely voluntary. A total of 411 students from various colleges within Kuwait University participated in the study. The inclusion criteria were undergraduate and graduate students aged 18–35 years. Participants outside this age range or those who failed to complete at least 50% of the survey were excluded. The sample size was calculated as, n = z^2^ pq/d^2^, where Z = (1.96)^2^, *p* = 0.5, q = 1 − *p* = 0.5, d = (0.05)^2^ [12], resulting in a minimum required sample size of 385 participants.

### 2.2. Survey Instrument

Data were collected using a structured, self-administered online survey developed in English and translated into Arabic. The survey was pre-tested among a small group of students (n = 10) to ensure clarity and cultural appropriateness. Revisions were made based on feedback before the final administration. Participants voluntarily completed an online self-administered survey via Google Forms. The survey instrument, adopted from the School of Public Health at Harvard University [13] and the Department of Health Behavior at Roswell Park Cancer Institute [14], underwent necessary modifications to suit Kuwait’s diverse student population. The survey included four sections: (1) demographic and anthropometric data, including age, gender, height, weight, academic major, and monthly personal spending budget (MPSB), defined as the total self-reported amount of money available to students for discretionary use each month. This included stipends provided by Kuwait University (typically capped at 200 KD), as well as any additional financial support from parents, sponsors, or part-time employment. Reported amounts were used to categorize students into spending brackets reflecting their disposable income available for food, transportation, and other living expenses; (2) frequency and types of SSB consumption; (3) awareness of SSB-related health risks (e.g., obesity, diabetes, cardiovascular diseases, and tooth decay); and (4) beverage preferences (natural unsweetened vs. sugar-sweetened fruit juices). Sugar intake from SSBs was estimated by asking participants to report their typical serving sizes and frequency of consumption for four beverage categories: regular soda, energy drinks, sports drinks, and sweetened iced tea. Sugar content per serving was determined based on nutrition labels of commonly available brands in Kuwait. Total daily sugar intake was calculated by multiplying the reported serving size and frequency of consumption by the sugar content.

### 2.3. Variables and Categorization

The primary exposure variables were frequency of SSB consumption and daily sugar intake, categorized as low (below the median) or high (above the median) based on Z-scores. The outcome variables included awareness of SSB-related health risks and preferences for unsweetened natural fruit juices. Demographic covariates included age (18–25 years and 26–35 years), gender, academic major (health sciences vs. non-health sciences), and MPSB (<50 KD, 50–100 KD, 101–200 KD, and >200 KD). BMI was calculated using self-reported height and weight and categorized as normal (≤25 kg/m^2^) or overweight/obese (>25 kg/m^2^).

### 2.4. Statistical Analysis

Data were analyzed using SPSS (version 28). Descriptive statistics (means, medians, frequencies, and percentages) were calculated for all variables. Chi-square tests assessed associations between SSB consumption frequency and demographic variables, as well as awareness of health risks. Logistic regression models estimated the odds ratios (ORs) for high sugar intake in relation to demographic and behavioral factors. Statistical significance was set at *p* < 0.05.

## 3. Results

### 3.1. Demographic and Anthropometric Characteristics

The study included 411 Kuwait University students, with females comprising the majority. Most participants were aged between 18 and 25 years. The mean BMI for the sample fell within the overweight range, with male students exhibiting significantly higher mean BMI compared to females. Approximately 61% of students were enrolled in non-health science disciplines. With regard to MPSB, nearly half of the sample reported having a budget between 101 and 200 Kuwaiti Dinars (KD), consistent with the student monthly stipend provided by Kuwait University. Full demographic details are summarized in Table 1.

### 3.2. Frequency of SSB Consumption

Regular soda was the most frequently consumed SSB, followed by sweetened iced tea, energy drinks, and sports drinks. The majority of participants reported consuming regular soda at least once per week, and a notable proportion reported daily intake. Weekly consumption rates of the other SSBs were considerably lower. The frequency distribution by beverage type is presented in Table 2.

### 3.3. Associations Between Frequency of SSB Consumption and the Sociodemographic Factors

SSB consumption frequency was significantly associated with gender, age, and field of study. Female students, those aged 18–25 years, and those enrolled in non-health science majors were more likely to report higher frequency of SSB consumption. Statistical analyses revealed significant associations for gender (*p* < 0.001), age group (*p* = 0.045), and academic discipline (*p* < 0.001), with a modest effect size observed for the association between field of study and consumption frequency (Cramer’s V = 0.21). No statistically significant associations were observed between SSB consumption frequency and BMI classification or MPSB. These patterns are depicted in Figure 1.

### 3.4. Behavior Towards SSB Consumption and Awareness of Health Risks

Overall, participants demonstrated high awareness of the health risks associated with frequent SSB consumption. More than 90% of students recognized the role of SSBs in contributing to obesity, type 2 diabetes, and tooth decay, while awareness of their association with cardiovascular, kidney, and liver diseases was also substantial. However, only 59% of respondents were able to correctly estimate the sugar content of SSBs, indicating a notable gap in specific knowledge. These findings are summarized in Figure 2.

### 3.5. Sugar Intake from SSBs

Sugar intake was calculated for 387 participants with complete consumption data. Regular soda accounted for the highest estimated median daily sugar intake, followed by energy drinks, sports drinks, and sweetened iced tea. Based on Z-score distributions, 41.6% of participants were categorized as high sugar consumers. Table 3 presents the intake levels across beverage types and the corresponding Z-score classification.

### 3.6. Associations Between Sugar Intake from SSBs and Gender, Age, BMI, Major Area of Study, MPSB, Awareness of Disease Risk, and Behavior Towards Sugary Drinks

Chi-square analyses revealed that high sugar intake from SSBs was significantly associated with male gender, younger age (18–25 years), and enrollment in non-health-related disciplines (*p* < 0.05 for all comparisons). No significant differences in sugar intake were observed by BMI classification or MPSB.

Regarding awareness, students who correctly identified the link between SSBs and obesity were significantly less likely to be high sugar consumers (*p* < 0.001). Similar associations were noted for awareness of the risks of type 2 diabetes (*p* = 0.02) and tooth decay (*p* = 0.03), while no significant associations were observed for awareness of cardiovascular, kidney, or non-alcoholic liver disease risks. Awareness of sugar content and preference for unsweetened juices were also inversely associated with high sugar consumption (*p* = 0.048 and *p* < 0.001, respectively). These associations are reported in Table 4A,B.

### 3.7. Regression Analysis of Sugar Intake from SSBs with Independent Variables and the Awareness of Disease Risk and Behavior Towards SSBs

Univariate logistic regression analyses demonstrated that students aware of the obesity risk associated with SSB consumption were approximately 86% less likely to report high sugar intake (OR = 0.142, 95% CI: 0.046–0.435; *p* < 0.001). Awareness of sugar content and preference for unsweetened juices were also independently associated with lower odds of high intake (OR = 0.653, *p* = 0.040; OR = 0.447, *p* < 0.001, respectively).

Multivariate logistic regression adjusting for age, gender, and field of study confirmed these associations. Awareness of obesity risk remained a significant protective factor (aOR = 0.167, 95% CI: 0.052–0.563; *p* = 0.003). Knowledge of sugar content (aOR = 0.653; *p* = 0.040) and preference for unsweetened juices (aOR = 0.498; *p* = 0.002) also continued to show significant inverse associations with high sugar intake. Male gender and enrollment in non-health disciplines were independently associated with a higher likelihood of excessive sugar consumption. Regression results are detailed in Table 5 and Table 6.

## 4. Discussion

This study assessed SSB consumption among Kuwait University students, estimated their daily sugar intake from these beverages, and evaluated their awareness of the associated health risks. It also explored the relationship between sugar intake from SSBs and various factors, including BMI (or weight status), MPSB, academic major, health awareness, and SSB consumption behavior.

The findings reveal distinct consumption patterns, with regular soda emerging as the most frequently consumed SSB. Specifically, 42% of students reported drinking soda 5–6 times per week, while 32% consumed it daily or more than seven times weekly. In contrast, energy drinks, sports drinks, and sweetened iced teas were consumed less frequently, with 21%, 9%, and 13% of students, respectively, reporting intake of seven or more times per week. These results align with previous research identifying soda as a major source of added sugars among university students and other populations worldwide [15,16,17,18,19,20,21]. For instance, Malik et al. [21] reported daily SSB intake among 54% of adolescents across 53 countries, and similar prevalence has been documented in the Middle East and U.S. settings [18,22,23]. This high consumption rate reflects a broader global pattern and underscores the need for targeted strategies to reduce SSB intake during this formative life stage.

This study revealed that participants’ median daily sugar intake from regular soda was 38 g [IQR: 14.04, 56.29], with 87% reporting regular consumption. This level exceeds the American Heart Association’s recommended daily limit for added sugar, which is 24 g for adult women and 36 g for men [24] and constitutes a substantial portion of the 50 g total daily limit recommended by the Dietary Guidelines for Americans 2020–2025 [25]. As such, little room remains for added sugars from other dietary sources like snacks and coffee beverages. These findings echo local estimates from The Medical City and SAMA Medical Services in Kuwait, which also reported high soda-related sugar intake [26]. However, unlike those surveys, this study observed lower sugar contributions from energy drinks (11.49 g), sports drinks (9.59 g), and iced tea (5.37 g), possibly due to differing consumption frequencies.

Interestingly, the relatively low energy drink consumption in this study may be influenced by the predominantly female sample, as male students are typically more frequent energy drink users [27,28]. These beverages are often marketed for their performance-enhancing effects and are popular among youth for staying alert during academic and social activities [29]. However, energy drinks often contain excessive caffeine and sugar, as well as additives such as taurine and ginseng, which can pose health risks including heart palpitations, insomnia, and digestive issues [30,31,32,33,34,35]. Public health messaging must therefore balance awareness of these risks with education on healthier alternatives.

Similarly, while sports drinks are designed to replenish electrolytes after physical exertion, their high sugar content raises health concerns [36]. In this study, sports drinks contributed approximately 13 g of sugar daily among high consumers. Prior research links overconsumption to obesity, metabolic disease, and oral health issues [35,37,38,39], suggesting these beverages should be consumed only during intense physical activity or under medical supervision [40].

Although black and green teas offer documented health benefits, these are often lost in commercial sweetened iced teas, which contain high levels of added sugars, artificial flavorings, and colorings [41,42,43]. Misconceptions about the healthfulness of iced teas may lead consumers to view them as better alternatives to soda, when in fact their sugar content is comparable. In the U.S., up to 80% of tea consumed in 2023 was sweetened iced tea [43], and similar patterns are seen in Kuwait, where such beverages often contain 19–26 g of added sugar per serving. In the current study, 33% of students reported weekly iced tea consumption, highlighting a need to correct public perception.

Sugar intake from SSBs in this study exceeded that reported in similar studies elsewhere. For example, research on college females at Saint John’s University, Minnesota, USA, showed average added sugar intakes of 10–15 g per day [44], while studies in Ankara and China reported daily sugar intakes from SSBs ranging from 11.3 to 13.4 g [45,46]. This discrepancy underscores the elevated sugar burden among students in Kuwait.

Research on U.S. adults further contextualizes the issue: approximately 30% exceed the recommended limit of 10% of daily calories from added sugar, with average intakes of 68 g (17 teaspoons) per day [47,48]. These figures align with this study’s findings and emphasize the broader public health implications of high added sugar consumption.

A key finding in the current study was that students who preferred unsweetened natural fruit juices were 55% less likely to consume high levels of sugar from SSBs. This suggests that promoting healthier beverage preferences may be an effective strategy to reduce total sugar intake. However, it is important to acknowledge that even unsweetened fruit juices are considered sources of free sugars according to World Health Organization (WHO) guidelines, which recommend limiting intake of free sugars—including those naturally present in fruit juices—to less than 10% of total energy intake, with further benefits observed below 5% [24]. While such juices may be a preferable alternative to carbonated SSBs, excessive consumption could still contribute substantially to daily sugar intake and should be moderated within public health messaging. The beverage industry has also responded to growing health concerns by implementing strategies such as direct sugar reduction, multi-sensory integration, and the use of sweeteners and enhancers [49]. These innovations could support broader interventions to shift consumption patterns among youth.

Gender and age also played important roles. While female students reported a higher frequency of SSB intake, male students consumed greater overall quantities, suggesting larger portion sizes among males. This may reflect greater concern about body image among females, who might choose smaller portions despite frequent consumption, a hypothesis supported by previous research [50,51]. Age-related differences were also evident, with students aged 26–35 reporting significantly lower sugar intake than their younger peers, potentially reflecting growing health awareness with age.

Although some previous studies have identified a positive association between SSB consumption and obesity [14], no significant link was found in the current study. The mean BMI of participants was in the overweight range (25.44 ± 6.88), with females averaging a normal BMI (24.63 ± 5.31) and males averaging 28.73 ± 10.66, classifying them as overweight. The lack of an association may be due to self-reporting bias, metabolic variability, or unmeasured confounders.

The study revealed high levels of awareness about the health risks of SSBs: 95% of students acknowledged the risk of obesity, 94% recognized the link to type 2 diabetes and tooth decay, and more than 85% identified associations with heart, kidney, and liver diseases. Knowledge of sugar content was also associated with reduced sugar intake, echoing findings from earlier studies [14,20,52,53,54].

However, although awareness was inversely associated with SSB intake, the direction of causality remains unclear. While greater awareness may influence consumption behavior, it is also possible that health-conscious individuals are more likely to seek out such information. Moreover, awareness alone may not always lead to behavioral change, especially when convenience, taste, affordability, or social norms come into play. For example, students may continue to consume SSBs despite knowing the risks if healthier alternatives are unavailable or less appealing. The cross-sectional nature of this study limits causal inference, and future longitudinal or intervention-based studies are needed to clarify whether awareness truly drives behavior change or is a marker of other underlying influences.

The study also found that students from non-health disciplines had significantly higher SSB consumption and sugar intake than those in health-related majors. This finding is consistent with previous research showing lower SSB intake among students in healthcare fields, likely due to their greater exposure to nutrition education [13,16]. The pattern underscores the importance of incorporating basic nutritional awareness into curricula across all academic disciplines, not just health sciences.

These findings highlight the critical role of early education and awareness in shaping dietary behaviors. Educational campaigns targeting university students, particularly those outside health-related fields, could help reduce sugar consumption and mitigate risks of obesity, type 2 diabetes, and other non-communicable diseases. Integrating nutrition education into school curricula and promoting access to healthier beverage options are essential steps toward improving dietary habits in this key population.

## 5. Strengths and Limitations

This study is the first to examine SSB consumption and related awareness among university students in Kuwait, providing a valuable contribution to the public health literature in this population. By quantifying sugar intake from SSBs rather than relying solely on frequency measures, the study offers a more detailed and nuanced understanding of consumption patterns. Additionally, it evaluates students’ awareness of the health risks associated with SSB intake, addressing a critical gap in existing research. The inclusion of students from various colleges within Kuwait University enhances the diversity of the sample, thereby improving the generalizability of the findings to the wider student population. Despite these strengths, certain limitations must be acknowledged. The cross-sectional design restricts the ability to infer causality between awareness and SSB consumption. Longitudinal or intervention-based studies would be better suited to explore the temporal sequence and causal relationships between knowledge and behavior. Moreover, reliance on self-reported data may introduce social desirability and recall bias, potentially leading to underestimation or overestimation of SSB intake. Additionally, while the study focused on university students, expanding future research to encompass other demographic groups, including younger adolescents and older adults, would provide a more comprehensive understanding of SSB consumption across age and socioeconomic strata. Such broader insight would support the development of more targeted and inclusive public health strategies.

## 6. Implications for Public Health and Future Research Prospects

There is a clear and urgent need for educational campaigns that raise awareness about the sugar content of SSBs and their associated health risks. Integrating these campaigns into university programs offers a strategic opportunity to reach young adults, a demographic critical for establishing lifelong dietary habits. Promoting healthier beverage alternatives, particularly natural and unsweetened options, could contribute meaningfully to reducing SSB intake in this population. Future research should prioritize the development and evaluation of intervention strategies specifically tailored to university settings. Particular attention should be given to students in non-health disciplines, who were shown to have higher levels of sugar intake. In addition, further investigations should explore the broader contextual and psychosocial factors contributing to high SSB consumption among young adults. These may include social influences, cultural norms, marketing exposure, taste preferences, and availability of healthier alternatives. Understanding these factors will be essential for designing multifaceted interventions that are both effective and sustainable.

## 7. Conclusions

This study highlights the high levels of sugar intake from SSBs among university students in Kuwait and underscores the pivotal role of health awareness in shaping dietary choices. The findings lay the groundwork for future research and targeted interventions aimed at reducing SSB consumption and promoting healthier eating behaviors among young adults. Enhancing awareness of the health risks associated with SSBs can serve as a catalyst for behavioral change and inform the design of public health initiatives. Addressing existing knowledge gaps and examining the underlying factors that influence SSB consumption will be crucial for the development of effective public health strategies. In the face of Kuwait’s rising obesity prevalence, targeted efforts to curb SSB intake should form a central pillar of national strategies to combat diet-related non-communicable diseases. The findings of this study should inform the design of culturally tailored educational campaigns and regulatory measures aimed at reducing sugar consumption and preventing the onset of diet-related chronic diseases among Kuwaiti youth.

## Figures and Tables

**Figure 1 nutrients-17-01646-f001:**
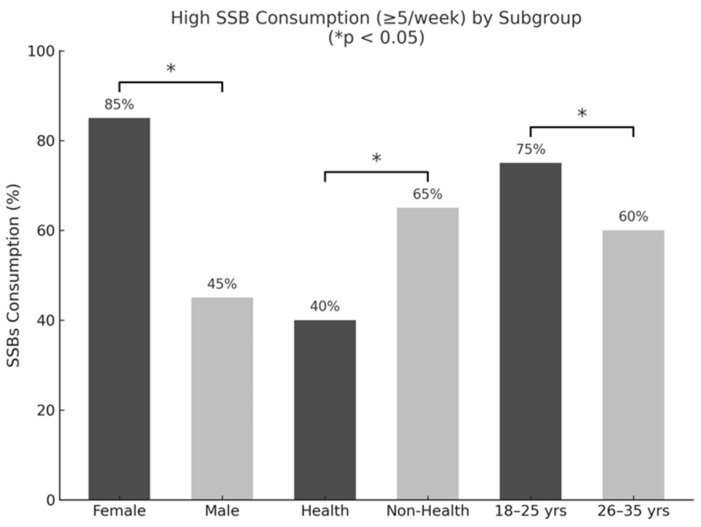
Frequency of sugar-sweetened beverage (SSB) consumption by gender, academic major, and age group among university students in Kuwait. Note: Prevalence of high sugar-sweetened beverage (SSB) consumption (≥5 times/week) among university students by gender, academic major (health vs. non-health sciences), and age group. Bars represent the percentage of participants within each subgroup reporting high-frequency SSB intake. Asterisks (*) denote statistically significant differences between subgroups (*p* < 0.05, chi-square test).

**Figure 2 nutrients-17-01646-f002:**
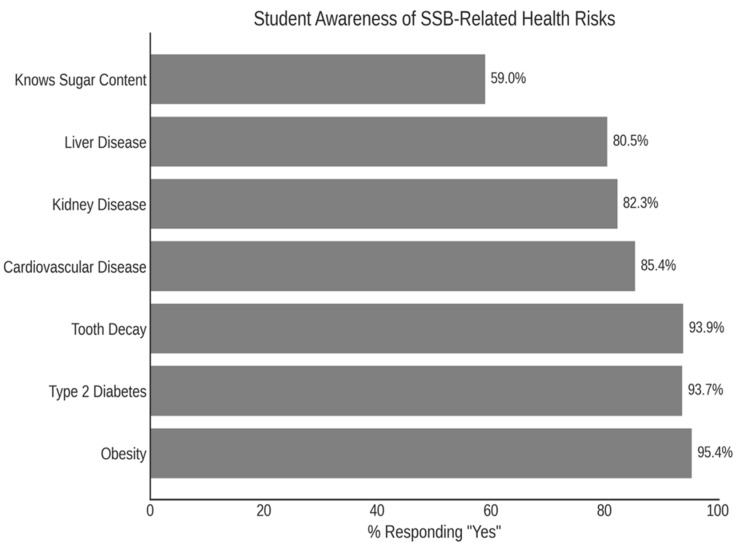
Awareness of health risks associated with sugar-sweetened beverage (SSB) consumption among university students in Kuwait. Note: Bars represent the percentage of students who responded “Yes” when asked whether SSBs are linked to specific health outcomes, including obesity, type 2 diabetes, tooth decay, cardiovascular disease, kidney disease, liver disease, and knowledge of sugar content in soft drinks.

**Table 1 nutrients-17-01646-t001:** Demographic characteristics of the study participants.

Variables	Female n = 331 (81%)	Male n = 80 (19%)	Total n = 411 (100%)
Age, n (%)			
18–25	262 (79.2)	50 (62.5)	312 (75.9)
26–35	69 (20.8)	30 (37.5)	99 (24.1)
Anthropometrics, mean ± SD			
Height ^a^, cm	158.75 ± 8.42	174.34 ± 6.67	161.80 ± 10.19
Weight ^b^, kg	62.13 ± 14.99	87.02 ± 30.69	66.99 ± 21.45
BMI ^c^, kg/m^2^	24.63 ± 5.31	28.73 ± 10.66	25.44 ± 6.88
Academic Major, n (%)			
Health sciences	147 (44.4)	12 (15.0)	159 (38.7)
Non-health sciences	184 (55.6)	68 (85.0)	252 (61.3)
MPSB ^d,e^, n (%)			
<50–100 KD	82 (25.2)	18 (23.7)	100 (24.9)
101–200 KD	167 (51.2)	27 (35.5)	194 (48.3)
201–300 KD	54 (16.6)	24 (31.6)	78 (19.4)
301 KD and more	23 (7.1)	7 (9.2)	30 (7.5)
BMI ^f^, (kg/m^2^) n (%)			
≤25	194 (59.0)	32 (40.0)	226 (55.3)
>25	135 (41.0)	48 (60.0)	183 (44.7)

Missing data: ^a^ n = 2, ^b^ n = 1, ^c^ n = 2, ^d^ n = 9, ^e^ MPSB = monthly personal spending budget, and ^f^ n = 2.

**Table 2 nutrients-17-01646-t002:** SSB consumption frequency by SSB type.

SSBs	≤1/Month n (%)	≤1/Week n (%)	2–4/Week n (%)	5–6/Week n (%)	≥7/Week n (%)	Total n (%)	Total n (%)
	Non-Consumers *	Consumers **					
Regular soda ^a^	115 (28)	102 (35)	66 (23)	29 (10)	92 (32)	289 (71.5)	404 (98.3)
Energy drinks ^b^	291 (72)	67 (58)	20 (17)	5 (4)	23 (21)	115 (28.3)	406 (98.8)
Sports drinks ^c^	359 (89)	26 (60)	9 (21)	4 (10)	4 (9)	43 (10.7)	402 (97.8)
Sweetened iced tea ^d^	273 (67)	83 (61)	26 (19)	9 (7)	17 (13)	135 (33.1)	408 (99.3)

* Non-consumers are the participants who may consume SSBs once or less in a month. The percentage is calculated based on the total sample of the specific SSB. ** SSB consumers = n (%) calculated based on the total sample minus the non-consumers of the specific SSB. Missing data for each SSB: ^a^ n = 7, ^b^ n = 5, ^c^ n = 9, ^d^ n = 3.

**Table 3 nutrients-17-01646-t003:** Daily sugar consumed (g) from SSB type and Z-scores of low sugar intake versus high sugar intake (n = 387).

Daily Sugar Consumed (g) by SSB Type		
SSB (n, %)	Median	IQR [25%, 75%]
Regular Soda (337, 87)	38.10	[14.04, 56.29]
Energy Drinks (164, 42.4)	11.49	[4.66, 15.47]
Sports Drinks (66, 16.9)	9.59	[4.26, 13.66]
Sweet Iced Tea (192, 49.7)	5.37	[3.71, 6.46]
Daily Total Sugar Intake (Z-Scores)		
High intake (n = 160, 41.6%)	0.70	[0.38, 1.36]
Low intake (n = 226, 58.4%)	−0.58	[−0.87, −0.32]
Z-Score from all SSBs (n = 387, 100%)	−0.24	[−0.68, 0.48]

**Table 4 nutrients-17-01646-t004:** Associations of sugar intake from SSBs with demographic variables, awareness of sugar content, disease risk, and preference for natural juices.

A: Associations of Sugar Intake from SSBs with Demographic Variables			
Variables	Low Intake	High Intake	χ^2^, *p*, Cramer’s V
Gender, n (%)			
Females	218 (68.3)	101 (31.7)	14.25, <0.001, 0.19
Males	30 (44.1)	38 (55.9)	
Age (years), n (%)			
18–25	183 (61.0)	117 (39.0)	4.01, <0.04, 0.04
26–35	65 (74.7)	22 (25.3)	
BMI (kg/m^2^), n (%)			
≤25	143 (66.8)	71 (33.2)	1.48, 0.23, 0.06
>25	104 (60.8)	69 (39.2)	
Academic Major, n (%)			
Health sciences	114 (77.6)	33 (22.4)	16.64, <0.001, 0.21
Non-health sciences	134 (55.8)	106 (44.2)	
MPSB ^a^, KD ^b^, n (%)			
<50–100	62 (70.5)	26 (29.5)	2.05, 0.56, 0.07
101–200	115 (63.2)	67 (36.8)	
200–300	39 (59.1)	27 (40.9)	
301 and above	13 (72.2)	5 (27.8)	
B: Associations of Sugar Intake from the SSBs with Disease Risk Variables			
Variables	Low Intake	High Intake	χ^2^, *p*, Cramer’s V
Obesity, n (%)			
Yes	238 (63.5)	137 (36.5)	15.27, <0.001, 0.184
No	5 (41.7)	7 (58.3)	
T2DM ^c^, n (%)			
Yes	234 (62.9)	138 (37.1)	2.10, 0.020, 0.124
No	6 (40.0)	9 (60.0)	
CVD ^d^, n (%)			
Yes	211 (62.2)	128 (37.8)	2.06, 0.563, 0.033
No	28 (58.3)	20 (41.7)	
KD ^e^, n (%)			
Yes	211 (63.0)	124 (37.0)	1.74, 0.532, 0.050
No	31 (59.6)	21 (40.4)	
NAFLD ^f^, n (%)			
Yes	185 (62.2)	113 (37.9)	0.20, 0.532, 0.018
No	54 (60.7)	35 (39.3)	
TD ^g^, n (%)			
Yes	235 (63.2)	137 (36.8)	4.25, 0.03, 0.134
No	5 (33.3)	10 (66.7)	
Aware of Sugar Content of SSBs, n (%)			
Yes	157 (68.3)	73 (31.7)	4.25, 0.048, 0.039
No	90 (57.3)	67 (42.7)	
Preference for Unsweetened Juices, n (%)			
Yes	167 (69.0)	75 (31.0)	14.88, <0.001, 0.192
No	73 (50.3)	72 (49.7)	

^a^ MPSB = monthly personal spending budget, ^b^ KD = Kuwaiti Dinar, ^c^ T2DM = Type 2 diabetes mellitus, ^d^ CVD = cardiovascular diseases, ^e^ KD = kidney disease, ^f^ NAFLD = non-alcoholic fatty liver disease, ^g^ TD = tooth decay.

**Table 5 nutrients-17-01646-t005:** Univariate logistic regression analysis with the binary dependent variable: low versus high sugar consumption from SSBs.

Dependent Variable, High Sugar from SSBs = 1	B	S.E.	Wald	*df*	Sig.	Exp (B)	95% CI		Nagelkerke R Square
							Lower	Upper	
Can frequent intake of SSBs increase the risk of obesity?	−1.95	0.57	11.65	1	<0.001	0.142	0.046	0.435	0.144
Can frequent intake of SSBs increase the risk of T2DM?	0.39	0.55	0.49	1	0.486	1.470	0.497	4.351	0.022
Can frequent intake of SSBs increase the risk of CVD?	−0.05	0.35	0.02	1	0.883	0.949	0.477	1.892	0.012
Can frequent intake of SSBs increase the risk of KD?	−0.19	0.36	0.26	1	0.608	0.830	0.408	1.691	0.001
Can frequent intake of SSBs increase the risk of NAFLD?	0.27	0.29	0.85	1	0.357	1.304	0.741	2.293	0.007
Can frequent intake of SSBs increase the risk of TD?	−0.54	0.47	1.34	1	0.247	0.580	0.231	1.458	0.013
Do you know how much sugar is in SSBs?	−0.43	0.21	4.23	1	0.040	0.653	0.435	0.980	0.106
Do you prefer sweetened fruit juices?	−0.81	0.22	14.63	1	<0.001	0.447	0.295	0.675	0.143

**Table 6 nutrients-17-01646-t006:** Multinomial logistic regression analysis with the binary dependent variable: low versus high sugar consumption from SSBs.

Dependent Variable: High Sugar from SSBs = 1	B	S.E.	Wald	*df*	Sig.	Exp (B)	95% CI		Nagelkerke R Square
							Lower	Upper	
Can frequent intake of SSBs increase of risk of obesity?	−1.79	0.59	9.06	1	0.003	0.167	0.052	0.536	0.144
Age = 1	−0.78	0.27	8.07	1	0.005	0.460	0.269	0.786	
Gender = 1	0.87	0.28	9.95	1	0.002	2.378	1.388	4.074	
Non-health sciences = 1	0.77	0.236	10.71	1	0.001	2.165	1.363	3.439	
Do you know how much sugar is in SSBs?	−0.43	0.21	4.23	1	0.040	0.653	0.435	0.980	0.106
Age = 1	−0.79	0.27	8.49	1	0.004	0.455	0.268	0.733	
Gender = 1	0.88	0.27	10.59	1	0.001	2.412	1.420	4.098	
Non-health sciences = 1	0.83	0.23	12.63	1	<0.001	2.293	1.451	3.624	
Your preference for fruit juices?	−0.69	0.22	10.02	1	0.002	0.498	0.323	0.767	0.118
Age = 1	−0.74	0.27	7.25	1	0.007	0.478	0.279	0.818	
Gender = 1	0.89	0.28	10.65	1	0.001	2.456	1.432	4.212	
Non-health sciences = 1	0.74	0.24	9.62	1	0.002	2.087	1.311	3.322	

Awareness of disease risks, awareness of sugar content of SSBs, and the preference for sweetened fruit juices are dichotomous questions with options of “No, Yes”, no = reference. Age: 18–25 = 0, 26–35 = 1; gender: female = 0, male = 1; academic major: health sciences = 0, non-health sciences = 1.

## Data Availability

The original data are presented in the article. The supplementary data is provided in Appendix A. Further inquiries may be directed to the corresponding author.

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
