# Peer review of "Sugar-Sweetened Beverage Consumption and Associated Health Risks Awareness Among University Students in Kuwait: A Cross-Sectional Study"

_nutrients, 2025, doi:10.3390/nu17101646_

Round 1
Reviewer 1 Report
Comments and Suggestions for Authors
While the topic is interesting, the study has some weaknesses.
The introduction should present the need for this study and the unique selling points of this study. For example, why should readers read this study? What is the contribution of this study compared to existing research?
I think the methodology needs more information about the survey process, for example, when was the survey conducted? What incentives were offered? What were the criteria for sample selection?
The authors presented their results separately for two groups of consumers. Is the difference between the groups statistically significant?
The authors should add a more detailed explanation of their findings.
Author Response
Reiewer 1:
Comment 1. Introduction lacks clarity on study’s unique contribution
Response: A concluding paragraph was added to the Introduction to clarify the study’s originality and public health relevance in Kuwait.
Change Location: Page 3, Lines 99-102.
Comment 2. Methodology requires additional detail on data collection
Response: The Methods section now specifies that the survey was conducted between May and July 2022, that no incentives were provided, and that participation was voluntary. Inclusion and exclusion criteria have also been clarified.
Change Location: Page 4, Lines 108-115.
Comment 3. Clarify whether differences between consumption groups were statistically tested.
Response: Chi-square tests and regression models are described in the Methods and applied in Results to examine group differences. Statistical significance is clearly reported.
Change Location: Results Section, Page 5, lines 154-158; Pages 9–12; Tables 4A and 4B.
Comment 4. Add more detailed explanation of findings.
Response: The Discussion section was expanded to interpret the major findings in light of previous research, including cultural and behavioral context.
Change Location: Pages 15–19, Lines 303–350.

Reviewer 2 Report
Comments and Suggestions for Authors
This is a well written manuscript highlighting a health pattern in Kuwait. I was surprise to learn the obesity levels in the country. Illustrates the importance of identifying risk factors, in this case SSB
Only a few minor edits
L56: having lifelong and life in the same sentence may be grammatically incorrect. Ideally, the word or version of word isn't written twice
What is allowance money? That may need to be operationally defined. An "allowance" in the United States is money that a parent gives to a child on a weekly or monthly basis (maybe associated with doing chores, etc.). Is it suppose to be the amount of money as part of one's budget?
Bottom of table 1. data (not date)
Table 2: format the n associated with >7 week to below to be similar to other columns
Figure 1: look for all "allowance money" locations. Does this need to be revised to a different term?
Author Response
Reviewer 2 Comments
Comment 1. Line 56: Redundant phrasing (“lifelong” and “life”)
Response: The sentence was reworded to remove redundancy and improve clarity.
Change Location: Page 3, Line 92-93.
Comment 2. Define “allowance money”.
Response: The term “monthly personal spending budget” has replaced “allowance money” throughout the manuscript. A formal operational definition was added to the Survey Instrument section.
Change Location: Page 4, Lines 128–133.
Comment 3. Table 1 footnote: Correct “date” to “data”
Response: Corrected as requested.
Change Location: Page 5, Table 1 footnote.
Comment 4. Table 2: Inconsistent formatting for “≥7/week” category.
Response: Formatting of frequency columns was corrected for alignment.
Change Location: Page 6, Table 2.
Comment 5. Figure 1: Ensure consistent terminology for “allowance money”
Response: All references in the figure and its caption were updated to “monthly personal spending budget (MPSB).”
See response to comment 2
Round 2
Reviewer 1 Report
Comments and Suggestions for Authors
The revision is great. Thus, the current draft is now acceptable. Well done.